# Diagnosis and Improvement of Combustion Characteristics of Methanol Miniature Reciprocating Piston Internal Combustion Engine

**DOI:** 10.3390/mi11010096

**Published:** 2020-01-16

**Authors:** Gangzhi Tang, Shuaibin Wang, Li Zhang, Huichao Shang

**Affiliations:** 1College of Mechatronics & Automotive Engineering, Chongqing Jiaotong University, Chongqing 400074, China; 622190940008@mails.cqjtu.edu.cn; 2College of Automotive Engineering, Chongqing University, Chongqing 400044, China; 20100701041@cqu.edu.cn (L.Z.); tanggz1980@163.com (H.S.)

**Keywords:** micro-energy power systems, micro internal combustion engine, combustion diagnosis, methanol

## Abstract

A micro-reciprocating piston internal combustion engine with liquid hydrocarbon fuel has the potential to supply ultrahigh density energy to micro electro mechanical system because of its high-density energy, simple structure, and mature energy conversion principle. However, the diagnostic test of the combustion characteristics of the micro reciprocating piston internal combustion engine shows that its combustion characteristics are poor, and the combustion rate was lower with the combustion duration of more than 50 °CA. The mean indicated pressure (Pmi) value was only 0.137 MPa, the combustion stability was very poor, and the cycle variation rate of the Pmi was up to 60%. To improve its combustion performance, the method to enhance combustion in micro-space is explored then. Mechanism studies have shown that the pyrolysis reaction of nitromethane and hydrogen peroxide can produce amounts of free radicals OH, with the possibility of improving the combustion of methanol. Therefore, a method for adjusting the composition of methanol fuel to enhance combustion is proposed, and the method is theoretically confirmed. Finally, based on this method, the test was carried out. The results showed that the combustion rate increased and the combustion duration decreased by 6% after adding nitromethane. The power performance was enhanced, and the Pmi value was increased by 30%. The combustion stability was enhanced, and the cycle variation rate of the Pmi was reduced to 16.9%. Nitromethane has a significant effect on improving the combustion characteristics of methanol, and the enhancement of the latter was mainly reflected in the ignition phase of the combustion process. This study indicates that exploring the fuel additive that can increase the concentration of OH radical in the reaction is an effective method to improve the micro-space combustion, which will facilitate the development of micro-piston internal combustion engine to supply energy to a micro electro mechanical system.

## 1. Introduction

Micro-energy power systems can achieve ultra-high energy density power output at the micro/intermediate scale by utilizing the energy of chemical combustion reaction to meet the wide needs of micro-aircraft, portable equipment, and other fields [1]. From the perspectives of energy density and conversion efficiency, micro-energy power systems with the liquid hydrocarbon fuel have the potential to compete with the lithium sulfur dioxide (LiSO_2_) battery system [2] (the lower calorific value of hydrocarbon fuel can reach 10^5^ kJ/kg level, while the LiSO_2_ battery’s energy density was only 10^2^ kJ/kg magnitude).

Representative research of micro-energy power systems includes micro-turbine engines [3], micro-triangular rotor Wankel engines [4], micro-steam turbines [5], miniature free piston engines [6], micro-thermal photovoltaic systems [7], micro-thermoelectric systems [8], and micro fuel cell systems [9]. Although various micro-energy systems have their own characteristics and advantages, there are important unresolved defects under the existing technical conditions. The high-precision processing of full 3D blade structure of micro-turbine engines and micro-steam turbines is hard to be realized. The problem of leakage of the micro-rotor engine has not been resolved. Thermoelectric conversion efficiency of micro-thermal power systems and photovoltaic conversion efficiency of micro-photovoltaic systems are not ideal. Micro-fuel cells also need breakthroughs in key components such as electrode catalysts and electrolyte films. Therefore, they are still in the theoretical confirmation research stage, and there is no mature commercial application.

The low calorific value of hydrocarbon fuel can reach to 10^5^ kJ/kg magnitudes. Therefore, even if its conversion efficiency is lower, it still has a high energy density. In addition, the principle of thermal conversion is mature; theoretically, it has the thermal efficiency of conventional macro-powerplants and has power performance with ultra-high energy density. In addition, it has many advantages, such as continuous energy supply and convenient refueling. Compared to the complex 3D blade structure of micro-turbine engine, the structure of the micro-reciprocating piston internal combustion engine is simpler and more adapted to miniaturized manufacturing [10]. A micro-reciprocating piston internal combustion engine with liquid hydrocarbon fuel has the possibility to supply ultrahigh density energy to a micro electro mechanical system for long-distance transmission.

Micro-space combustion is the primary domain of research on micro-energy power systems. The study on the micro-space combustion has shown that micro-size effects such as high surface-to-volume ratio and short residence time deteriorate the fluctuation in the combustion cycle in the micro-space, incomplete combustion, miss-fire, and other abnormal combustion phenomena [11,12,13]. To solve the above problems, research on micro-space combustion shows a trend of development towards surface catalytic combustion, which can make full use of the feature of high surface-to-volume ratio in micro space [14,15]. However, for the micro reciprocating piston internal combustion engine with a variable surface, surface catalytic combustion cannot make full use of its high surface-to-volume ratio feature because the main combustion process happens near the top dead center of piston. Moreover, its catalytic efficiency may be affected by the drastic change combustion temperature and pressure. Therefore, how to achieve stable and efficient transient combustion in micro-space becomes a key challenge that must be solved in the development of a micro reciprocating piston internal combustion engine energy system to supply energy to micro electro mechanical systems.

The micro combustion engine has characteristics such as a large relative loss of heat, short fuel residence time, and small space of combustion chamber. High-efficiency and fast combustion must be achieved to ensure the power performance of micro-internal combustion engines. Thus, the fuel with the characteristic of sufficient and fast combustion is needed. Methanol, which is clean and efficient, has been used as a newly alternative fuel for conventional engines. The flame propagation rate of the methanol’s combustion is 32.7 cm/s, which is the largest after the hydrogen and acetylene and 1.5 times that of gasoline. Compared to the petroleum-based fuel, methanol contains a small number of carbon atoms, a small molecular weight. The energy required for ignition is only 0.140 MJ, so that the methanol is a fuel with low ignition energy. Methanol has a large oxygen content, and carbon granules won’t exist. Liquid fuel methanol is easy to store, transport, and fill. Christian et al. indicated that NO can promote methanol oxidation [16,17]. Zhang et al. indicated that the high-temperature decomposition of nitromethane will produce amounts of NO intermediate components [18]. Nitromethane combustion releases high heat (708.1 Kg/mol).

In view of this, a combustion test platform is established for the miniature reciprocating piston internal combustion engine to test the basic combustion characteristics of the methanol fuel glow ignition of platinum wire in micro-space. Furthermore, the methods to enhance the combustion characteristics of methanol fuel in micro space are explored. It is proposed and tested to add nitromethane and hydrogen peroxide to promote methanol combustion to improve combustion performance of a methanol micro-piston internal combustion engine. This method is convenient and there is no need for additional equipment compared to surface catalytic combustion. The study is helpful to the development of the ultra-high energy density power system of the micro piston internal combustion engine.

## 2. Combustion Characteristics Test of Methanol Fuel in Micro-Space

A combustion test platform for a miniature reciprocating piston internal combustion engine was constructed, as shown in Figure 1. The test system was driven by GRZ1500 high speed variable frequency motor (Shuangrenhe Electromechanical Equipment Co., Ltd., Dongguan, China), and the speed of engine was adjusted by the motor too. In addition, the speed of the motor was adjusted by a E300-2S0015 transducer (Shenzhen Sifang Electric Company, Shenzhen, China). The AHB-202 hysteresis brake (Changlin Automation Technology Co., Ltd., Wuxi, China) was used as an adjustable load device to absorb the output torque of the engine and the motor. The engine was a miniature air cooled two-stroke glow ignition of platinum wire reciprocating piston internal combustion engine. Its basic parameters can be seen from Table 1. A Kistler6052B quartz pressure sensor (Kistler, Winterthur, Switzerland) was adopted to collect the combustion pressure signal in the micro-piston internal combustion engine. The sensor has amounts of advantages such as a high sensitivity, wide measuring range, small thermal impact effect, a small load drift, and a high natural frequency. The parameters are shown in Table 2. The charge signal output by the piezoelectric sensor was converted into a voltage signal by a 5011B charge amplifier (Kistler, Winterthur, Switzerland). The data was analyzed by a DEWE2010 combustion analyzer (Dewetron, Graz, Austria). At the same time, the AVL365X angle instrument (AVL List GmbH, Graz, Austria) collects the crank angle signal, and the sampling resolution was set to be 0.2 °CA. The DM6801A type k thermocouple (Wuxi Guang Chen Longxing Industrial Control Instrument Co., Ltd., Wuxi, China) was adopted to investigate the thermal load of engine. The engine was cooled by a fan. The test condition was a full load at 6000 r/min.

Figure 2 shows the characteristic curve of the cylinder pressure and the combustion heat release rate in a micro piston internal combustion engine under the working condition of a full load at 6000 r/min. Figure 2a shows the instantaneous heat release curve. Figure 2b shows the cumulative heat release curve. The fuel is pure methanol; its lower calorfic value is 21.12 MJ/kg with a density of 0.81 g/cm^3^. It can be seen from Figure 2 that the cylinder pressure of the platinum wire glow ignition micro-piston internal combustion engine was relatively low when taking pure methanol as fuel, and the maximum combustion pressure Pmax was only 0.65 MPa. The lower combustion pressure and the delays of corresponding crank angle will reduce the expansion ratio of the engine and reduce the thermal efficiency of the cycle. The mean indicated pressure Pmi (indicated work/swept volume) at the test was at a low level, and the Pmi value obtained from the Indicator diagram was only 0.137 MPa.

At the same time, it can be seen from the instantaneous heat release rate in the cylinder that the combustion speed was lower during the combustion and exothermic process, and the combustion duration was longer, especially during the late heat release period. Here: according to the first law of thermodynamics, the instantaneous heat release rate (heat release of 1 kmole mixture per crank angle) is composed of the following three parts: increment of thermodynamic energy of mixture, work done by mixture, and heat dissipation. They were calculated according to actual engine parameters and measured pressure change with crank angle.

The crank angles corresponding to the burned mass fraction 5%, 10%, 50%, 90% (CA05, CA10, CA50, CA90) on the cumulative exothermic curve obtained from the instantaneous heat release rate are shown in Table 3. It can be seen from Table 3 that the combustion start (taking the cumulative heat release 5% corresponding to the crank angle as the combustion starting) was later, and the crank angle corresponding to CA05 was delayed to 12.4 °CA after top dead center (ATDC). Because the combustion heat release rate was lower, the combustion duration was longer, resulting in the crank angle corresponding to the cumulative heat release period from 10% to 90% (CA10 to CA90) was more than 50 °CA. Among them, the early stage of rapid combustion (CA10 to CA50) was up to 19.4 °CA, which extended about 10 °CA compared with the conventional models; in the late stage of rapid combustion (CA50 to CA90), it reached 30.7 °CA, which was about 15 °CA longer than the conventional models. In addition, the moments corresponding to 50% and 90% of the cumulative combustion heat release (CA50 and CA90) are relatively late, resulting in the combustion exotherm extending to the late expansion phase, causing incomplete combustion and decreasing thermal efficiency.

Figure 3 shows the variations in the mean indicated pressure Pmi and Pmax for continuous 120 test cycles when a pure methanol fuel was used. In general, the rate of cyclic variation should not be greater than 10%. It can be seen from the Figure 3 that the stability of combustion was very poor, and the rate of cyclic variation was higher. The cyclic variation rate (standard deviation/mean value*100%) of Pmi was tested to be about 60%, which indicates that the combustion of micro space in the cylinder was very unstable. Because the miss-fire cycle was greater and the combustion state was worse, the maximum combustion pressure Pmax has a larger fluctuation range. The Pmax cyclic variation rate was tested to be 36%.

Figure 4 was the variations in the crank angle corresponding to the cumulative heat release 5% and 90% (CA05 and CA90) for the 120 continuous test cycles under the working condition of full load of 6000 r/min. It can be seen from the Figure that the starting point of combustion (CA05) varies greatly. The variation ranges from 5 °CA before top dead center (BTDC) to 40 °CA ATDC, and its variation range was close to 50 °CA. The changes in the ignition time will lead to a higher cyclic variation; as the CA05 in the Figure 4 was greatly delayed, the corresponding Pmi value in Figure 3 declines more obviously. This shows that the instability of the combustion starting point was the main reason for the high cyclic variation in the combustion process in the micro-space.

The large variation of CA05 causes the crank angle corresponding to 90% (CA90) of cumulative combustion exotherm was drastic. It can be seen from Figure 4 that the variation range of CA90 was more than 50 °CA, leading to the change of the combustion duration (CA05 to CA90) being intensified, resulting in the combustion cyclic variation being greatly increased.

## 3. Effect of Additives on the Combustion Characteristics of the Platinum Wire Glow Ignition in Micro-Space

### 3.1. Study on the Promotion Mechanism of Nitromethane and Hydrogen Peroxide to Methanol Combustion

The chemical reaction mechanism of methanol combustion was analyzed. The chemical mechanism used here is a detailed chemical reaction mechanism of methanol combustion constructed by Li [19]. The mechanism that has been experimentally verified from low temperature to high temperature is comprehensive. During the combustion of methanol, methanol undergoes a series of dehydrogenation processes and oxidation processes to generate CO, and then produces the final product CO_2_ through reaction with OH radicals. The main reaction pathway was shown in Figure 5. Among them, OH was the most important free radical in the reaction. In theory, the combustion of methanol can be promoted by increasing the amount of OH as well as other hydroxyl radicals.

The chemical reaction kinetics analysis shows that the strong oxidant hydrogen peroxide will produce a large number of OH through reaction H_2_O_2_ (+M) = OH + OH (+M). The Nitromethane pyrolysis reaction products, CH_3_ and NO_2_, will consume mainly through CH_3_ + HO_2_ = CH_3_O + OH, NO_2_ + H = NO + OH and NO + HO_2_ = NO_2_ + OH under certain conditions. These reactions above will transform HO_2_ into a more active free radical OH. Therefore, it was proposed to add nitromethane and hydrogen peroxide to enhance the combustion characteristics in the micro space of methanol fuel.

The combustion kinetics of methanol and hydrogen peroxide mixed fuel was analyzed. The chemical mechanism used here is a detailed chemical reaction mechanism of methanol combustion constructed by Li [19]. Li constructed a detailed chemical reaction mechanism for methanol combustion based on summarizing the mechanisms that appeared in the past, and Li has verified the mechanism by the way of experiments from low temperature to high temperature. The mechanisms that are comprehensive can predict methanol combustion in a wider range. The combustion process simulation was performed by using a closed homogeneous reactor model provided in the Chemkin-PRO program. Figure 6 was the effect of different hydrogen peroxide ratios on ignition delay when the equivalent ratio Φ = 1; the initial pressure is 3 MPa. It was obvious that, when the methanol fuel was added with hydrogen peroxide, its ignition delay was shorter than that of the pure methanol; the greater the proportion of the increase, the more favorable for the ignition.

Figure 7 shows combustion analysis sensitivity of methanol and methanol–hydrogen peroxide hybrid fuel by using the closed homogenous reactor model provided in the Chemkin-PRO program (Version 4.5, Ansys Inc., Canonsburg, PA, USA). Here, the equivalent ratio is 1, initial temperature and pressure are respectively 800 K and 3 MPa, and the mixture contains 10% volume of hydrogen peroxide. The Figure 7 shows the eight elementary reactions with the largest temperature coefficients. As shown in the Figure, for pure methanol, sensitivity analysis shows that the elementary reaction with the maximum positive temperature sensitivity coefficient was CH_3_OH + O_2_ = CH_2_OH + HO_2_, and the second one is CH_3_OH + HO_2_ = CH_2_OH + H_2_O_2_. Here, positive temperature sensitivity coefficient means that the elementary reaction promotes the temperature increase, and promotes the combustion. After the addition of hydrogen peroxide, the elementary reaction with the maximum positive temperature sensitivity coefficient was H_2_O_2_ (+M) = OH + OH (+M). The reaction of CH_3_OH + O_2_ = CH_2_OH + HO_2_ has no significant effect on temperature. It could be seen that H_2_O_2_ produces OH through reaction H_2_O_2_ (+M) = OH + OH (+M). The addition of hydrogen peroxide directly added the active radical OH into the whole reaction, shortening the chemical reaction chain and accelerating the combustion.

In order to analyze the effect of nitromethane on methanol combustion, the chemical reaction mechanism of methanol and nitromethane mixed fuel combustion was constructed [17,18,19,20,21,22,23], including the sub mechanism of methanol and nitromethane combustion and the oxidation reaction sub-mechanism of methanol and nitrogen oxide, with a total of 52 components and 240 elementary reactions. (The mechanism can be seen in Appendix A.) The mechanism has been verified by the engine combustion test. Figure 8 was the ignition delay of different nitromethane adding ratios when the equivalence ratio was 1, and the initial pressure is 3 MPa. Obviously, the addition of nitromethane will reduce the ignition delay of methanol, which was favorable for ignition.

In order to further analyze the effect mechanism of the nitromethane on the combustion of methanol, the reaction path of the main components of methanol and nitromethane mixed fuel combustion was analyzed, which was shown as Figure 9. Figure 9 shows that the main reactions of methanol are as follows: CH_3_OH is consumed primarily by dehydrogenation reactions with hydrogen–oxygen free radicals and oxygen, as well as NO_X_. Among them, the most methanol is consumed in the reaction with OH. There is about 42% methanol consumed in the reaction of CH_3_OH + OH = CH_2_OH + H_2_O, 29% methanol was consumed in the reaction of CH_3_OH + OH = CH_3_O + H_2_O, and 19% methanol is consumed in the reaction of CH_3_OH + HO_2_ = CH_2_OH + H_2_O_2._ The above three reactions are the most important consumption routes for methanol. The rate of low temperature starting reaction of methanol (CH_3_OH + O_2_ = CH_2_OH + HO_2_) is very low, which consumed about 3% methanol. The dehydrogenation reaction between methanol and NO_X_ consumes less than 10% of methanol.

CH_2_OH is consumed by dehydrogenation reaction, and the vast majority of CH_2_OH is consumed by reaction CH_2_OH + O_2_ = CH_2_O + HO_2_. In addition to being generated by the dehydrogenation reaction of CH_3_OH, CH_3_O is also generated by reaction CH_3_ + NO_2_ = CH_3_O + NO and CH_3_ + HO_2_ = CH_3_O + OH. The vast majority of CH_3_O is consumed by high temperature decomposition reaction of CH_3_O + M = M + CH_2_O + H + M and dehydrogenation reaction of CH_3_O + O_2_ = CH_2_O + HO_2_. The main consumption reaction of CH_2_O is CH_2_O + OH = HCO + H_2_O. HCO generates CO mainly by way of the dehydrogenation reaction of HCO + O_2_ = CO + HO_2._ Most CO is oxidized by OH into CO_2_.

It also can be seen that the addition of nitromethane has little effect on the main reaction pathway of methanol, but it adds many important branched chain reactions to make the reaction more complicated. The main reaction pathway of nitromethane was its decomposition reaction of CH_3_NO_2_ (+M) = CH_3_ + NO_2_ (+M), which was also the main source of CH_3_ and NO_2_, while the majority of CH_3_ was consumed by the reaction of CH_3_ + HO_2_ = CH_3_O + OH, resulting in OH during the reaction; NO_2_ was mainly consumed by the reaction of NO_2_ + H = NO + OH and CH_3_ + NO_2_ = CH_3_O + NO, which was also the main production reaction of NO; the generated NO was consumed through the reaction of NO + HO_2_ = NO_2_ + OH and NO + HCO = HNO + CO, during which OH was produced; until then, the nitromethane was finally integrated into the main reaction pathway of methanol.

Thus, nitromethane generated CH_3_ and NO_2_ through its own decomposition reaction, and generated a lot more active OH through the above reaction. In addition, at the same time, in the process, the generated NO and NO_2_ with stronger activity will be involved in the dehydrogenation reaction of methanol and its dehydrogenation intermediate products, making the whole chain reaction various, thereby enhancing the combustion of methanol.

### 3.2. Test of Methanol Combustion Enhanced by Nitromethane and Hydrogen Peroxide

In order to test the effect of nitromethane and hydrogen peroxide on the combustion characteristics of a micro piston internal combustion engine of methanol with the glow ignition of platinum wire, two kinds of mixed fuels of methanol/nitromethane and methanol/hydrogen peroxide were prepared for combustion diagnosis, respectively. Among them, the volumetric ratio of nitromethane, hydrogen peroxide, and mixed fuel was 10%, the purity of nitromethane was 99.99%, and hydrogen peroxide was analytically pure; its mass fraction was 30%.

After adding nitromethane and hydrogen peroxide, the measured cylinder pressure and instantaneous heat release rate of the micro-piston internal combustion engine are shown in Figure 10. As it can be seen from the Figure, after adding nitromethane, the cylinder pressure significantly increased, the heat release rate increased significantly too, and the combustion duration was short. It was calculated from the indicator diagram that, after adding nitromethane, the mean indicated pressure increased significantly. Its *Pmi* value increased from 0.137 MPa at pure methanol to 0.182 MPa at present, with an increase of 30%. However, after adding hydrogen peroxide solution, the cylinder combustion conditions worsen, and the cylinder pressure reduced, the combustion heat release rate slowed, and the combustion duration extended.

In addition, after adding nitromethane, the onset of combustion was significantly earlier, with the corresponding CA05 obtained from the cumulative heat release rate advanced to 8.5 °CA ATDC, as shown in Table 4, which was about 4 °CA ahead of pure methanol, and the combustion duration (CA05 to CA90) was shortened by about 3 °CA. However, after the addition of hydrogen peroxide solution, the combustion situation in the cylinder became worse, the ignition time was delayed, the combustion heat release rate slowed down, and the combustion duration was prolonged. The crank angle corresponding CA05 obtained from the cumulative release rate was delayed to 24.7 °CA ATDC, and the crank angle corresponding CA90 was delayed to 78.5CA ATDC. The combustion duration (CA05 to CA90) was increased to 54 °CA. The mean indicated pressure *Pmi* value measured drop to 0.078 MPa with a large decrease rate. 

It could be seen that the effect of hydrogen peroxide solution on the combustion of methanol in micro space was poor, which was different from the theoretical analysis of the previous. The cause was analyzed as follows: it may be because the mass fraction of water in the hydrogen peroxide solution was as high as 70%, the presence of amounts of aqueous solutions counteracts the effect of the increase in the mole fraction of OH. In addition, it is assumed to have the same initial conditions in the theoretical analysis, but, in the actual process, there are differences in the properties of different fuels. When the liquid phase is changed to a gas phase, the temperature of the mixture will change. For hydrogen peroxide with low mass fraction, there is too much water. The latent heat of vaporization of water (2257.2 kJ/kg) is about twice as much as methanol. The injection of hydrogen peroxide solution makes the temperature of the initial mixture lower, resulting in a negative impact on the combustion.

When two kinds of mixed fuels are used, the change in the mean indicated pressure *Pmi* of the 120 continuous test cycles was shown in Figure 11. It can be seen from the Figure that, compared with the no nitromethane fuel, after adding nitromethane, the range of *Pmi* value tends to concentrate and the number of misfire cycles decreases.

Among them, the *Pmi* cycle variation rate of being fueled with methanol/nitromethane decreased from 60% to 16.9% when no nitromethane was used. It could be seen that the effect of the nitromethane additive on the combustion cycle fluctuations in the glow ignition mode of the platinum wire in the micro piston internal combustion engine was obvious. However, after adding the additive of hydrogen peroxide, the number of misfire cycles increased, and the *Pmi* cycle variation rate increased to 116%.

Figure 12 shows the correlation among the mean indicated pressure *Pmi*, the duration of combustion, and the starting point of combustion CA05 when methanol and two mixed fuels are burned. As pure methanol, the change range of CA05 was large, which was more than 40 °CA. The ignition time was relatively delayed, and there are more cycles in which CA05 appears after 25 °CA ATDC, resulting in a serious misfire, *Pmi* decreased rapidly. After adding the nitromethane additive, the range of CA05 variation was narrowed and CA05 concentrated between 0 °CA ATDC and 20 °CA ATDC, resulting in a lower misfire rate and enhanced combustion stability. CA05 showed an early trend, with a significant reduction in the number of cycles in which CA05 appears after 25 °CA ATDC. After the hydrogen peroxide solution was added, the ignition time was severely delayed, and the number of cycles in which CA05 appears after 30 °CA ATDC increased rapidly, resulting in a serious misfire.

It can be seen from Figure 12 that the change of combustion duration had a strong correlation with CA05. With the delay of CA05, the duration of combustion significantly prolonged. Only when CA05 appeared after 25 °CA ATDC was there a superficial phenomenon of a rapid decrease in combustion duration due to misfire. After adding nitromethane, CA05 was ahead of time, and the duration of the combustion had a shortening trend, which was conducive to the reduction in the cyclical changes.

From those above analyses, it could be seen that the combustion-supporting characteristics of nitromethane additive in micro-space were much better than that of hydrogen peroxide. Nitromethane can promote CA05 in advance, and shorten the duration of combustion, lower the cycle change, and enhance the power performance. However, after adding the hydrogen peroxide solution, the combustion rate decreased, the combustion stability deteriorated, the misfire cycle increased, and it did not show the expected combustion-supporting effect.

## 4. Effect of Nitromethane with Different Proportions on the Methanol Combustion Characteristics in Micro-Space

The tests on combustion of the methanol added with different proportions of nitromethane were conducted to evaluate its combustion-supporting effect on micro-space combustion. Among them, the volume ratio of nitromethane during the test was 0%, 5%, 10%, and 15%, respectively.

The measured combustion cylinder pressure and instantaneous heat release rate of methanol with different proportions of nitromethane were shown in Figure 13. It can be seen from the Figure that, as the proportion of nitromethane increased, the maximum cylinder pressure increased significantly, with the highest cylinder pressure at a nitromethane ratio of 15%. With the increase of the ratio of nitromethane, the maximum heat release rate increased, and the duration of combustion also shortened. Especially, the shortening at the later stage of combustion exotherm was more obvious. Among them, when the nitromethane ratio was 15%, the instantaneous heat release rate was the greatest.

It was calculated from the cumulative heat release rate that, when different proportions of nitromethane-methanol were in combustion, the crank angles corresponding to 5%, 10%, 50%, and 90% of the burned quality mass fraction are shown in Table 5. As can be seen from the Table 5, with the proportion of nitromethane increased, the starting point of combustion showed an earlier trend. Nitromethane had effectively promoted combustion, so that the starting point of combustion showed an earlier trend.

It was found from the test that the promoting effect of nitromethane additives on the methanol combustion in micro-space with the way of glow ignition of platinum wire was positively correlated with the proportion of nitromethane added. In Table 5, methanol fuel had the earliest ignition time at a nitromethane addition rate of 15% and its ignition point was 5.3 °CA ATDC, which was about 7 °CA earlier than when no nitromethane was fired, and the corresponding duration of combustion (CA05 to CA90) shortened from 52.8 °CA to 44.9 °CA and shortened by about 8 °CA.

In addition, its corresponding mean indicated pressure *Pmi* increased with the increase in nitromethane ratio. Among them, when the nitromethane ratio was 15%, its maximum *Pmi* value was 0.193 MPa, which increased about 40% compared with in non-nitromethane.

Table 6 was the cyclic variations in the value of mixed fule combustion maximum pressure *Pmax* and the mean indicated pressure *Pmi* with different proportions of nitromethane. The *σ*_Pmax_ was the standard deviation, the $\overset{-}{p}$_max_ was the mean value, and the *CoV*_Pmax_ was the cycle variation rate. It can be seen from the Table 6 that, when the nitromethane ratio was 15%, the cycle change of mixed fuel combustion was the lowest, the cycle variation rate of *Pmi* was reduced to about 9.4%, and the cycle variation rate of *Pmax* was reduced to about 19.5%. With the increasing of nitromethane, the $\overset{-}{p}$_mi_ and $\overset{-}{p}$_max_ both increase, and these standard deviations both reduce. Nitromethane can promote distribution of the combustion cycle.

## 5. Conclusions

Research in view of the combustion characteristics of the glow ignition of platinum wire of the miniature reciprocating piston internal combustion engine was conducted, a method of adjusting the fuel composition to enhance the combustion characteristics of the micro-space was proposed. The specific conclusions are as follows:

(1) Burning pure methanol fuel, the combustion rate of the glow ignition of platinum wire of the miniature internal combustion engine was relatively low, and its combustion duration was longer, with the duration of more than 50 °CA. The cylinder pressure was lower, and its *Pmi* value was only 0.137 MPa. The combustion stability was poor, and the cycle variation rate of *Pmi* was up to 60%. Its combustion characteristics are poor.

(2) The mechanism research shows that nitromethane can produce a lot of highly active free radicals OH through a series of key reactions at high temperature, and the hydrogen peroxide can also produce a large number of free radicals OH through pyrolysis reaction, which has the possibility of enhancing methanol combustion. Therefore, a method for adjusting the composition of methanol fuel to enhance combustion has been proposed. Theoretical studies confirmed that nitromethane and hydrogen peroxide can enhance the combustion of methanol in micro-space.

(3) Experimental results show that, after adding nitromethane, the combustion rate increases, and the duration of combustion shortens by 6%. The cylinder pressure was obviously increased, and its power performance was improved. The *Pmi* value was increased by 30%. The combustion stability was improved, and the cycle variation rate of *Pmi* was decreased to 16.9%. The enhancing effect of nitromethane on the combustion characteristics of methanol in micro space was obvious. In addition, it has a positive relationship between the enhancing effect of the combustion and the nitromethane proportion.

(4) The starting point of combustion has a good correlation with the mean indicated pressure and the combustion duration. Nitromethane influences the mean indicated pressure and its cyclic variation by adjusting the distribution of combustion onset. Nitromethane has a positive impact on the methanol combustion process in micro-space, which was mainly reflected in the ignition phase during the combustion process.

(5) This study explores and proposes the method to promote micro-space combustion of a micro-piston internal combustion engine. It indicates an important research standpoint, exploring the fuel additive that can increase the concentration of OH radical in the reaction, which is an effective method to improve the micro-space combustion. The tested maximum output power of the micro-piston internal combustion engine reaches 80 W. The study will facilitate the development of the ultra-high density energy micro-piston internal combustion engine to supply energy to the micro electro mechanical system.

## Figures and Tables

**Figure 1 micromachines-11-00096-f001:**
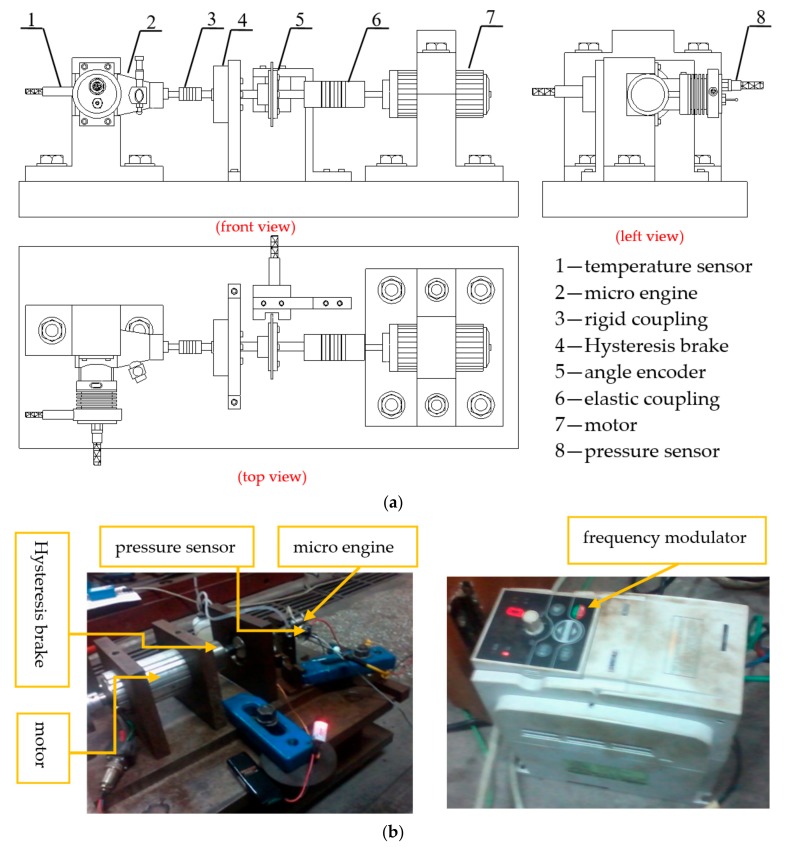
Experimental platform of the micro internal combustion engine (**a**) experimental platform; (**b**) experimental installation.

**Figure 2 micromachines-11-00096-f002:**
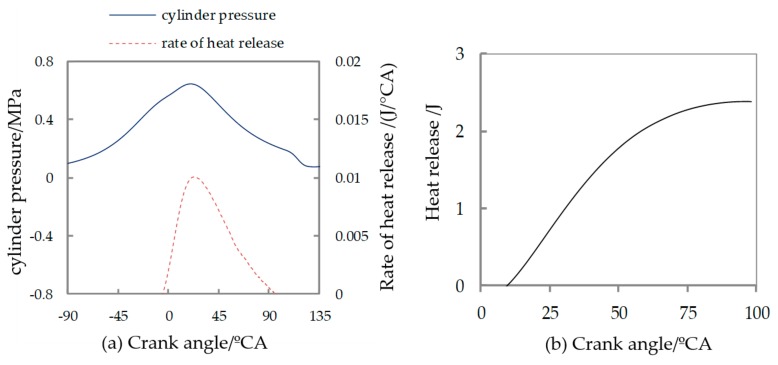
(**a**) Cylinder pressure and (**b**) heat release for methanol fuel.

**Figure 3 micromachines-11-00096-f003:**
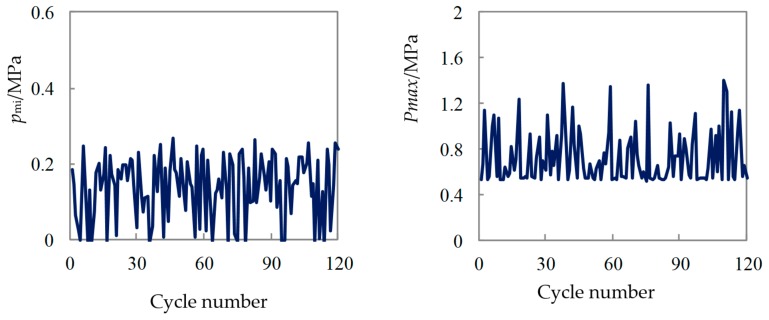
Cyclic variations of Pmi and Pmax for methanol fuel.

**Figure 4 micromachines-11-00096-f004:**
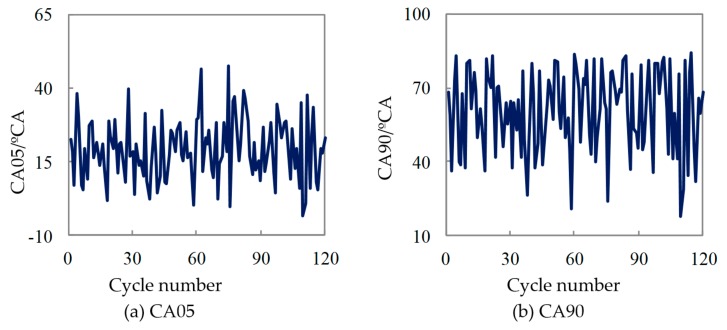
Variation of CA05, CA90 for methanol fuel.

**Figure 5 micromachines-11-00096-f005:**
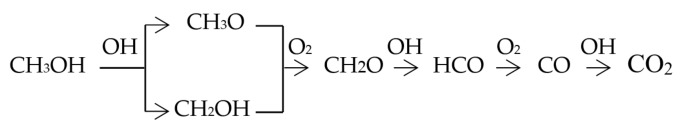
Combustion reaction scheme of methanol fuel.

**Figure 6 micromachines-11-00096-f006:**
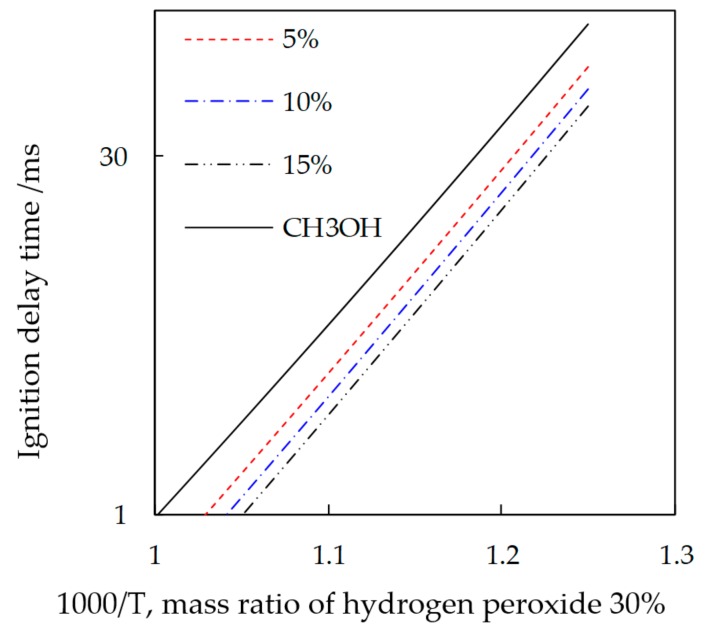
Effect of hydrogen peroxide on methanol combustion.

**Figure 7 micromachines-11-00096-f007:**
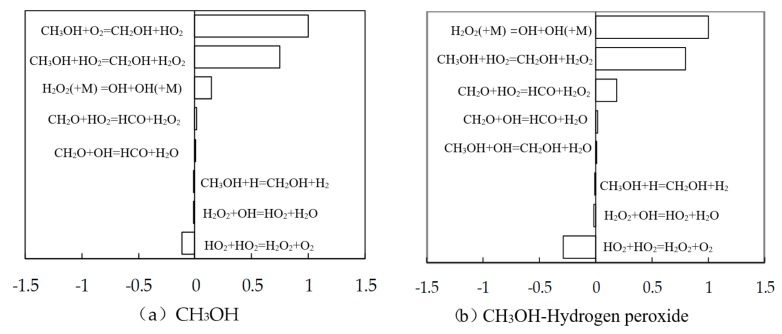
Combustion analysis sensitivity of methanol and methanol–hydrogen peroxide hybrid fuel.

**Figure 8 micromachines-11-00096-f008:**
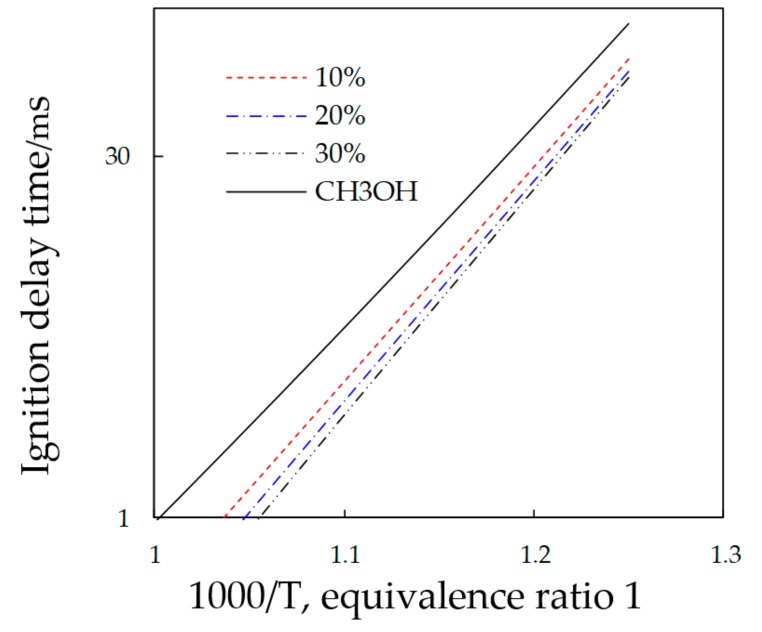
Effect of nitromethane on methanol combustion.

**Figure 9 micromachines-11-00096-f009:**
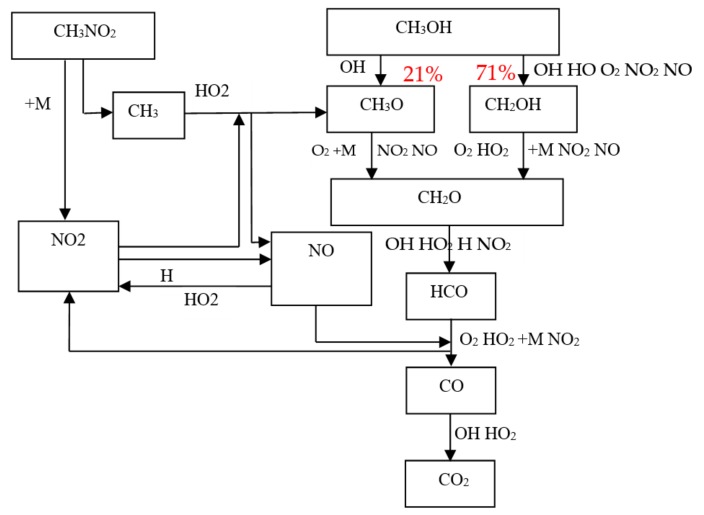
Elementary reaction pathways of methanol and nitromethane mixed fuel.

**Figure 10 micromachines-11-00096-f010:**
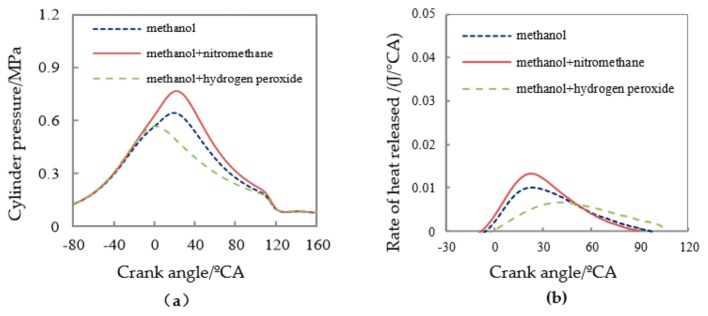
(**a**) Variations of cylinder pressure with different additives; (**b**) variations of the rate of heat release with different additives.

**Figure 11 micromachines-11-00096-f011:**
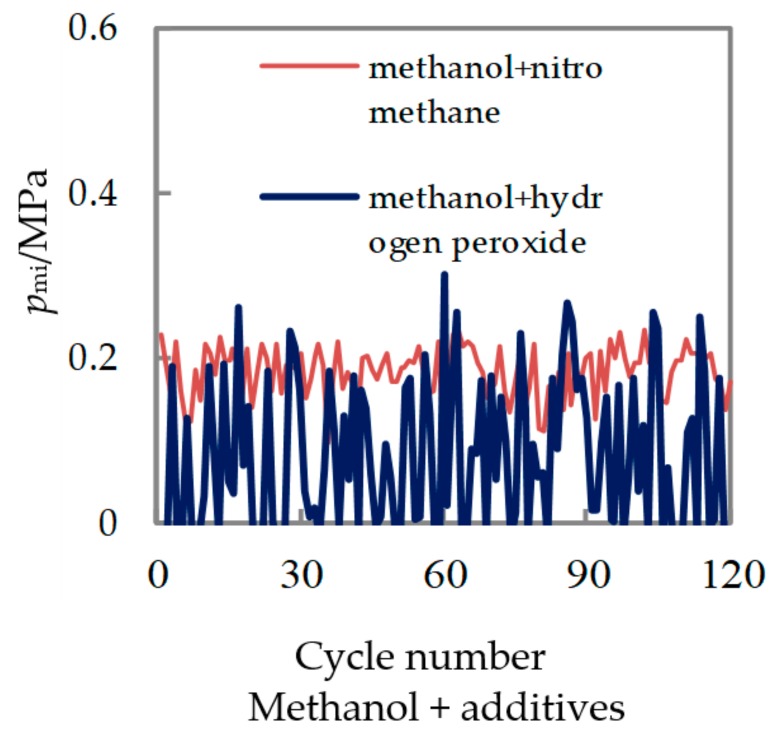
Cyclic variations of Pmi with different additives.

**Figure 12 micromachines-11-00096-f012:**
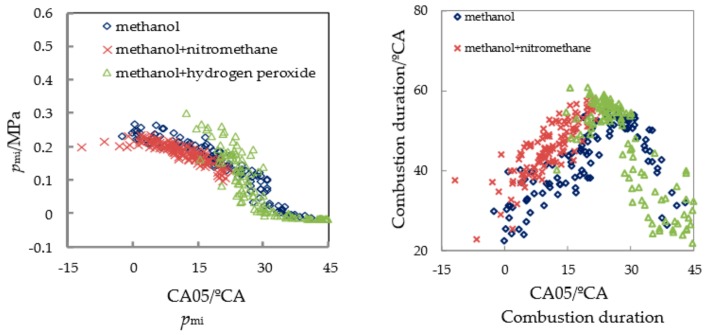
Pmi and combustion duration correlated with CA05 with different additives.

**Figure 13 micromachines-11-00096-f013:**
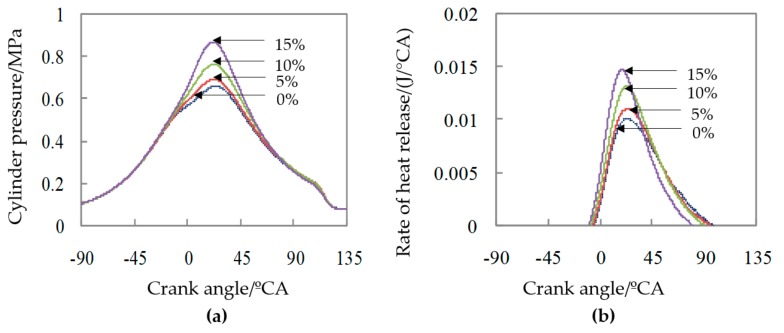
(**a**) Effect of nitromethane on the cylinder pressure; (**b**) Effect of nitromethane on the rate of heat release.

**Table 1 micromachines-11-00096-t001:** The basic parameters of engines.

Displacement/L	Cylinder Bore/mm	Stroke/mm	Compression Ratio
0.001	11.25	10	8

**Table 2 micromachines-11-00096-t002:** The technical parameters of a 6052B pressure sensor.

Parameters	Argument Value
Pressure range/bar	0–250
Limit value/bar	300
Natural frequency/kHz	130
Operating temperature range/°C	−50–400
Linear error/%FSO	≤±0.4
Sensitivity/pC/bar	16
Thermal impact error/bar	≤±0.5
Impact-resistant/g	2000
Capacity/PF	5
Install threads	M5 × 0.5

**Table 3 micromachines-11-00096-t003:** Characteristic parameters of methanol fuel.

Fuel	Pmax/MPa	APmax/°CA	Pmi/MPa	CA05/°CA	CA10/°CA	CA50/°CA	CA90/°CA
methanol	0.65	19.2	0.137	12.4	15.1	34.5	65.2

**Table 4 micromachines-11-00096-t004:** Characteristic parameters of methanol fuel with additives.

Fuel	*P*max/MPa	*AP*max/°CA	*Pmi*/MPa	CA05/°CA	CA10/°CA	CA50/°CA	CA90/°CA
methanol/10% nitromethane	0.767	22.6	0.182	8.5	11.5	29.7	58.2
methanol/10% hydrogen peroxide solution	0.569	1.0	0.078	24.7	27.8	50.2	78.5

**Table 5 micromachines-11-00096-t005:** Characteristic parameters with different proportion of nitromethane for methanol.

Ratio of Nitromethane/%	*Pmax*/MPa	A*Pmax*/°CA	*Pmi*/MPa	CA05/°CA	CA10/°CA	CA50/°CA	CA90/°CA
0	0.646	19.2	0.137	12.4	15.1	34.5	65.2
5	0.687	20.4	0.153	11.1	13.8	32.7	61.7
10	0.767	22.6	0.182	8.5	11.5	29.7	58.2
15	0.874	21.4	0.193	5.3	8.0	24.3	50.2

**Table 6 micromachines-11-00096-t006:** Combustion cyclic variations of methanol-nitromethane fuel.

Ratio of Nitromethane/%	*Pmax*			*Pmi*		
	*σ*_Pmax_/MPa	$\overset{-}{p}$_max_/MPa	*CoV*_Pmax_/%	*σ*_Pmi_/MPa	$\overset{-}{p}$_mi_/MPa	*CoV*_Pmi_/%
0%	0.25	0.71	35.8	0.08	0.14	61.1
5%	0.20	0.72	27.9	0.05	0.15	31.8
10%	0.19	0.79	23.9	0.03	0.18	16.9
15%	0.17	0.89	19.5	0.02	0.19	9.4

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
