# Peer review of "Diagnosis and Improvement of Combustion Characteristics of Methanol Miniature Reciprocating Piston Internal Combustion Engine"

_micromachines, 2020, doi:10.3390/mi11010096_

Round 1

Reviewer 1 Report

In this paper, the authors hypothesis that including ignition improvers into the base fuel will increase performance. The approach is modeled in Chemkin prior to experiments. Hydrogen peroxide does not perform as expected and is likely due to the presence of a large amount of water. Nitromethane improves performance, but the authors cap the mix at 15% without commenting on limitations. The writing is readable, though it could use editing by a native English speaker. The topic area is one of great interest, though I do not think the results are surprising or overly novel.

Reviewer 2 Report

In the paper, methanol and Nitromethane/hydrogen peroxide have been considered as a fuel and methanol reactivity boosters in a confined size reciprocating combustion engine, respectively. To do so, the authors carried out several experimental and simulation studies. The results showed that adding Nitromethane to methanol could sensibly boost the mixture reactivity in terms of faster ignition delay time and rate of heat released. In this regard, the following comments would be suggested:

1- The motivation of the paper is not clear enough. Why did the authors try methanol as a fuel in such a small engine? However, Nitromethane is well known for RC applications.
2- According to limitations of the reciprocating small size engines for converting chemical energies to mechanical/electrical ones (e.g. huge losses and higher friction levels and low intrinsic thermal performances and etc.), why did authors investigate this type of energy convertors?

3- In Section 2, the presented explanations are vague and unclear. Which type of pressure sensors including type and uncertainty were applied and so on. Details of engine should be provided.

4- In figure 2, what is the difference between right- and left handside sub-figures? The figure is not clear. What is the meaning of the colors in the rigth handside sub-figure?

5- In figure 5, which chemical mechanism has been used to get the expression?

6- H2O2 (+M) = HO + HO (+M) is very important for temperature range <1200 K. What is the meaning of the high temperature in the paragraph?

7- According to zero-dimension modelling, why did not the authors apply a high fidelity comprehensive chemical mechanism instead of the simplified one and after that compare the permanence of high fidelity chemical mechnism with the simplified one to verify its performance under the studied conditions.

8- In figure 6, the Y-axis should be shown in log scale.

9- In page 6-pragraph 2, what is the meaning of positive coefficient? Is it promoting the reactivity? details of the sensitivity analysis should be provided.

10- In figure 6 and 7, the ignition delay time (IDT) performance of the applied mechanism should be evaluated using standard Nitromethane experimental IDTs or high fidelity chemical mechanism.

11- In page 6, "...the oxidation reaction sub-mechanism
262 of methanol and nitrogen oxide, with a total of 52 components and 240 elementary reactions..". From where? source of the mechanism should be presented?

12- In page 7-second paragraph, an accurate mechanism for methanol and NOx is required. So using a simplified mechanism dosen't seem to be acceptable in this regard.

13- In figure 8, details and H-abstraction percentages should be provided on the arrows and reaction fluxes.

14- What is the meaning of dehydrogenation? Does it H-abstraction?

15- In page 7, the third paragraph should be rewritten.

16- What is the meaning of "analytically pure"?

17- In pages 8 and 9, the presented reasons for the inverse effect of adding hydrogen peroxide on the mixture reactivity should be shown and verified properly. it could be confirmed by doing some computer and simulation tests.  

Round 2

Reviewer 2 Report

Now, the paper could be recommended for publication.